# Repositioning of Quinazolinedione-Based Compounds on Soluble Epoxide Hydrolase (sEH) through 3D Structure-Based Pharmacophore Model-Driven Investigation

**DOI:** 10.3390/molecules27123866

**Published:** 2022-06-16

**Authors:** Erica Gazzillo, Stefania Terracciano, Dafne Ruggiero, Marianna Potenza, Maria Giovanna Chini, Gianluigi Lauro, Katrin Fischer, Robert Klaus Hofstetter, Assunta Giordano, Oliver Werz, Ines Bruno, Giuseppe Bifulco

**Affiliations:** 1Department of Pharmacy, University of Salerno, Via Giovanni Paolo II 132, 84084 Fisciano, SA, Italy; egazzillo@unisa.it (E.G.); sterracciano@unisa.it (S.T.); druggiero@unisa.it (D.R.); glauro@unisa.it (G.L.); 2The Italian Foundation for Cancer Research, Institute of Molecular Oncology, Via Adamello 16, 20139 Milan, MI, Italy; marianna.potenza@ifom.eu; 3Department of Biosciences and Territory, University of Molise, C. da Fonte Lappone, 86090 Pesche, IS, Italy; mariagiovanna.chini@unimol.it; 4Department of Pharmaceutical/Medicinal Chemistry, Institute of Pharmacy, Friedrich-Schiller-University, Philosophenweg 14, 07743 Jena, Germany; katrin.fischer.1@uni-jena.de (K.F.); robert.klaus.hofstetter@uni-jena.de (R.K.H.); oliver.werz@uni-jena.de (O.W.); 5Institute of Biomolecular Chemistry (ICB), Consiglio Nazionale delle Ricerche (CNR), Via Campi Flegrei 34, 80078 Pozzuoli, NA, Italy; asgiordano@unisa.icb.cnr.it

**Keywords:** drug repositioning, soluble epoxide hydrolase, drug discovery, computational techniques, chemical synthesis, anti-inflammatory agents

## Abstract

The development of new bioactive compounds represents one of the main purposes of the drug discovery process. Various tools can be employed to identify new drug candidates against pharmacologically relevant biological targets, and the search for new approaches and methodologies often represents a critical issue. In this context, in silico drug repositioning procedures are required even more in order to re-evaluate compounds that already showed poor biological results against a specific biological target. 3D structure-based pharmacophoric models, usually built for specific targets to accelerate the identification of new promising compounds, can be employed for drug repositioning campaigns as well. In this work, an in-house library of 190 synthesized compounds was re-evaluated using a 3D structure-based pharmacophoric model developed on soluble epoxide hydrolase (sEH). Among the analyzed compounds, a small set of quinazolinedione-based molecules, originally selected from a virtual combinatorial library and showing poor results when preliminarily investigated against heat shock protein 90 (Hsp90), was successfully repositioned against sEH, accounting the related built 3D structure-based pharmacophoric model. The promising results here obtained highlight the reliability of this computational workflow for accelerating the drug discovery/repositioning processes.

## 1. Introduction

Computational techniques are valuable and stimulating tools useful for the identification of new potential drug candidates. In a typical drug discovery process, a large number of molecules are designed, selected/filtered out, synthesized, and biologically evaluated, in order to identify new promising bioactive compounds. This approach is time and cost consuming and often provides disappointing results [1]. In order to overcome this issue, drug repurposing computational-based strategies can be applied (Figure 1) [2,3]. Indeed, in silico methods represent excellent tools for the repositioning of different molecular platforms, including already approved drugs, natural products with unknown mechanisms, and newly synthesized compounds designed for a given target but not performing as expected.

In this work, we show the successful repositioning of a small set of compounds employing a 3D structure-based pharmacophore model-driven approach [4], which finally led to new inhibitors of soluble epoxide hydrolase (sEH). sEH, belonging to the arachidonic acid cascade and involved in inflammatory pathologies, represents an interesting target deeply investigated in the last years for the treatment of inflammation and related disorders. It is responsible for epoxyeicosatrienoic acids (EETs) degradation to the corresponding dihydroxyeicosatrienoic acids (DHETs), leading to the lack of biological benefits, such as anti-inflammatory, vasodilatory, anti-hypertensive, cardioprotective, and analgesic effects, mediated by EETs [5]. In this regard, the inhibition of sEH causes decreased plasma levels of pro-inflammatory cytokines and nitric oxide metabolites [6], in addition to increased lipoxin formation, supporting the resolution of inflammation [7]. These data suggest that sEH inhibitors may have valuable therapeutic effects in the treatment and management of inflammatory diseases [8]. In mammalian cells, different epoxide hydrolase isoforms have been identified, and each of them takes part in detoxifying mutagenic and carcinogenic xenobiotic oxiranes [9]. If related to other isoforms, e.g., microsomal epoxide hydrolase (mEH), the relative abundance of sEH in most tissues, such as liver [10], kidney [11], and intestine, leads to its major contribution in the metabolism of epoxy fatty acids in vivo [12].

Because of the significant benefits achievable with the blockage of sEH activity, various binders have been identified featuring the urea and amide groups representing the most popular and potent class of sEH inhibitors [13,14,15]. Moreover, among the already identified sEH inhibitors, a large number of compounds, both fragment and drug-like items, were co-crystallized with the enzyme, thus offering the possibility to provide insight into the binding mode and into the key interaction needed for the inhibition, which is useful for the design of novel potent bioactive compounds. On this basis, starting from a careful analysis and comparison of the structural data arising from a number of the above-mentioned protein/inhibitor co-crystal structures, we here developed a 3D structure-based pharmacophore model for sEH, representing a promising tool for drug design [16]. Actually, several pharmacophoric models have already been developed for sEH; specifically, a receptor-based pharmacophore model [17], a ligand-based pharmacophore model [18], and a 3D structure-based pharmacophore model, the latter obtained using only one ligand [19]. In the last few years, our research group has been involved in the discovery of novel soluble epoxide hydrolase inhibitors (sEHi) and, accordingly, the development of sEH 3D structure-based pharmacophore model represents a valuable strategy for accomplishing this aim [20,21]. Indeed, in addition to the identification of novel compounds with anti-inflammatory and anticancer activity targeting mPGES-1, representing another key target of our research interests [20,22,23,24], we also focused on other targets belonging to the arachidonic acid cascade to identify multitarget agents with greater benefits than single-target inhibition [25]. In light of these premises, the 3D structure-based pharmacophore model was developed by collecting the necessary spatial definitions from the specific coordinates of multiple co-crystallized inhibitors in the specific sEH binding site, obtaining a model directly placed in the pocket cavity of the enzyme, bearing the 3D information from multiple known co-crystallized inhibition. The developed 3D structure-based pharmacophore model was applied as a valuable tool for selecting new binders of this target and, specifically, it proved to be suitable not only for the identification of new sEHi, but also for drug repositioning strategy in order to re-evaluate a library of shelved compounds synthesized over the years featuring no promising results for the originally selected target. In this study, 190 different organic compounds originally designed and synthesized for different targets, i.e., mPGES-1, HSP90, BRD9, PARP, TANK1, JMJD3, HSF1, and BAG3, were submitted to a 3D pharmacophore-based repositioning investigation, and six quinazolinedione derivatives, belonging to the set of molecules initially designed as inhibitors of heat shock protein 90 (Hsp90) [26] but showing poor binding, were selected as novel promising sEH inhibitors.

## 2. Results and Discussion

The workflow aimed at the repositioning of 190 in-house synthesized compounds and leading to the quinazolinedione-based compounds on sEH, as new inhibitors endowed with anti-inflammatory properties, is reported in Figure 2.

Specifically, the reported workflow concerned the re-investigation of an in-house library of 190 organic synthesized compounds during the latest years for different targets, e.g., mPGES-1, HSP90, BRD9, PARP, TANK1, JMJD3, HSF1, BAG3 (SMILES of the library compounds are in Appendix A). 

During the computational repositioning campaign, quinazolinedione-based molecules were here selected among the 190 investigated items against sEH. These compounds were originally identified as putative Hsp90 inhibitors, and no binding was then detected against this target (see Section 3). In the following paragraphs, detailed information regarding the different related steps is described.

### 2.1. Original Building of the Library of Quinazolinedione-Based Compounds and Virtual Screening on Hsp90

The rationale for the choice of the quinazolinedione core for the development of novel potential Hsp90 inhibitors lies in the previous discovery by our research group of several Hsp90 inhibitors bearing this scaffold [27]. In order to further investigate and perform an optimization of the previously identified compounds, CombiGlide software (version 4.4, Schrödinger, Inc., New York, NY, USA) [28] (Schrödinger suite) was employed. In this way, a large quinazolinedione-based virtual library of synthesizable compounds was built, considering different items for the generation of three libraries endowed with 5, 6, or 7 carbon chains at N3, in order to evaluate the influence of the chain length on the biological activity. Furthermore, commercially available aromatic amines (2924) were combined with each selected scaffold (Figure 3). After applying different filters, based on pharmacokinetics properties, including the Lipinski’s rule of five, the obtained libraries were reduced to 3639 drug-like compounds as input for the molecular docking-based virtual screening on the C-terminal domain of Hsp90 [29,30] (see Section 3 and Appendix A).

After docking calculations, the most promising compounds were selected for the synthesis and the subsequent biophysical assays.

### 2.2. Chemical Synthesis

The synthesis (Figure 1) of the selected compounds **3**–**8** (Figure 4), based on a dihydroquinazoline scaffold, started from the reaction of isatoic anhydride (1*H*-3,1-benzoxazine-2,4-dione), a suitable synthetic building block (**1**) with aminopentanoic/esanoic/eptanoic acid (**a**–**c**) in the presence first of triethylamine and then of formic acid to obtain the intermediates **2a**–**2c**, respectively (Figure 1) [31,32].

The second step involved the amide bond formation using the diisopropylcarbodiimide (DIC) and 1-hydroxybenzotriazole (HOBt), between **2a**–**2c** and different heterocyclic amines (**d**–**g**). Following this synthetic procedure, compounds **3**–**8** were obtained in good yields (see Section 3) [33].

### 2.3. Biophysical Assays on HSP90 and Repositioning on Soluble Epoxide Hydrolase (sEH) through 3D Structure-Based Pharmacophore Model-Driven Investigation

The synthesized compounds **3**–**8** were then tested in a surface plasmon resonance-based assay to address their potential binding towards Hsp90 (see Section 3). However, none of the selected molecules showed a significant affinity for the protein. Analyzing these data retrospectively and considering our experience with the design and identification of Hsp90 modulators towards both N- [34,35] and middle/C terminal domain [27,29,36,37,38,39], we ascribed our negative results to the high conformational change degree associated with remarkable rearrangements in Hsp90 structure during its mechanism of action. Moreover, no crystal structure of the human active Hsp90 middle/C-terminus bound to the inhibitor was disclosed in a close and active state, which hampered a punctual and detailed structural-based drug design.

As above reported, we recently developed a 3D structure-based pharmacophoric model for sEH, since it represents a target of our interest (see Section 1), in order to facilitate the identification of possible anti-inflammatory and anticancer agents. It is important to note that the use of this specific computational tool led us to the successful identification of novel bromodomain-containing protein-9 (BRD9) inhibitors after developing pharmacophore models specifically built for this protein module [4]. In details, we here implemented a 3D structure-based pharmacophore model directly built in the binding site of sEH (X-ray protein structure with PDB code: 5AI5 [40]). Specifically, starting from 108 sEH ligand/protein co-crystal structures, whose coordinates are available in the Protein Data Bank, we firstly filtered out the crystallographic structures without ligands and those containing fragment-like compounds. In this way, 20 ligands, extracted from the related sEH co-crystal structures, were chosen, based on (a) the presence of the ureidic group [15] or its bioisosteres, fundamental for the interaction with the amino acids involved in the mechanism of action, i.e., Asp335, Tyr383, Tyr466; (b) similar binding mode; and (c) IC_50_ values in the low micromolar/nanomolar range. All of these criteria were set in order to provide a robust and reliable 3D pharmacophore model, reflecting the common characteristics of the most active inhibitors.

Considering these aspects, sEH ligand/protein co-crystal structures (Figure 5) (PDB codes: 1EK2, 1VJ5, 3ANS, 3ANT, 3WKE, 4HAI, 4OCZ, 4OD0, 5AI5, 5AK5, 5AKE, 5ALG, 5ALP, 5ALU, 5ALZ, 5AM1, 6AUM, 6FR2, 6HGX and 6YL4 [40,41,42,43,44,45,46,47,48,49,50]), were downloaded from www.rcsb.org (accessed on: 4 February 2021).

Following this approach, a 3D structure-based pharmacophore model featuring five points was developed in an sEH crystal protein structure (PDB code: 5AI5, chosen for a good resolution of 2.28 Å). Specifically, this model contains two H-bond acceptor features (named “A”), a hydrophobic function (named “H”), an aromatic moiety (named “R”), and an H-bond donor feature (named “D”) (AADHR pharmacophore model, which we called “pharm-sEH”, Figure 6 and Appendix A). Interestingly, the acceptor and donor functions, namely A1 and D1 in Figure 6, cover the typical urea moiety or its bioisosteres present in most sEH binders discovered so far, and they are placed close to the related key interacting residues Asp335, Tyr383, and Tyr466. In addition, the aromatic function R1 is related to the interaction with His524 via π–π stacking, which was indeed detected in a number of co-crystallized inhibitors. The other two functions, A2 and H1, completed this pattern.

The developed “pharm-sEH” represented a valuable computational tool for speeding up the identification of new putative sEH inhibitors. The 3D structure-based pharmacophore model not only represents the starting point for the design and identification of novel sEH inhibitors, but is a very versatile tool that can be successfully implemented for other purposes, such as drug repositioning campaigns, as in this work, applying the steps of the workflow reported in Figure 7. In particular, as previously mentioned, 190 items available in our laboratory were submitted to a drug repositioning campaign, in which quinazolinedione-based compounds **3**–**8** were included. All the synthesized compounds were preliminarily screened with the generated “pharm-sEH” pharmacophore model using the “Ligand and database screening” tool in Phase [63,64,65]. In this way, a conformational search aimed to assess a basic structure complementary with sEH binding site was performed. After this step, 89 compounds respecting all of the pharmacophoric points of “pharm-sEH”, were then submitted to molecular docking calculations. The obtained docking poses were further subjected to a more restrictive “in place” pharmacophore-based screening since the accounted “pharm-sEH” model was directly placed onto the sEH binding site. Interestingly, only the quinazolinedione-based compounds **3**, **4**, **6**, **7**, and **8** successfully passed all workflow steps. It is worth noting that **5** was the only one in this series that did not meet all the pharmacophoric points (4/5), although it was selected as well for the subsequent biological evaluation as “negative control” in order to corroborate the reliability of the developed “pharm-sEH” and to validate its applicability for accelerate the identification of new sEH inhibitors.

Docking poses related to five of the six accounted compounds matched all the five pharmacophoric points inside the protein counterpart, namely compounds **3**, **4**, **6, 7** and **8** (Figure 8). In Table 1, for each investigated compound, the following parameters are reported: (a) the number of matched features (i.e., Num Sites Matched in Table 1), (b) the PhaseScreen score, which indicates a measure of how well the molecule fits within the pharmacophoric model, and (c) the docking score, which indicates the extent of binding established between the ligand and protein counterpart.

All of the compounds **3**–**8** were tested by in vitro experiments against sEH, in order to corroborate the computational outcomes.

### 2.4. Biological Evaluation on sEH

Compounds **3**–**8** were screened against sEH at a concentration of 10 µM in order to evaluate the activity on this target in a cell-free assay (see Section 3). It is worth noting that sEH features two active domains, namely a C-terminal domain epoxide hydrolase and an N-terminal featuring lipid phosphatase activity. On the other hand, all of the calculations and the subsequent biological assays were consistently conducted considering the specific modulation of the activity of the C-terminal hydrolase domain. The results, which are means of triplicate experiments, showed that **3**, **4**, **6, 7** and **8** were able to interfere with sEH, reflecting a reduction of sEH activity (Table 2 and Figure 9A) compared to the DMSO used as vehicle control (100%). AUDA (100 nM) was used as positive control, which inhibited sEH as expected (data not shown). As expected, compound **5** did not show significant inhibition against sEH.

The obtained experimental outcomes (Table 2) corroborated the in silico predictions, highlighting the robustness and reliability of the “pharm-sEH” model. Above all, this tool can be conveniently employed to implement the repositioning campaigns since, as predicted, only the compounds matching all the pharmacophoric features showed a significant reduction of the activity of the enzyme.

In addition, for the most promising compounds, IC_50_ values were determined (8.8 ± 1.5 µM and 4.5 ± 1.0 µM for **3** and **4**, respectively, Figure 9B). Moreover, to evaluate the toxicity profile of the investigated compounds (**3**–**8**), MTT assays were performed and, accordingly, all compounds were not cytotoxic, thus representing promising drug candidates (Figure 9C). Interestingly, both compounds **3** and **4** contain the 1,4-benzodioxin substituent, which is essential for matching the pharmacophoric features (see Figure 8) and the establishment of key amino acid interactions (Figure 10). Remarkably, the known inhibitor R4N (see PDB code: 5ALG, IC_50_ = 30.0 nM) features the same chemical group, suggesting that it could represent a good starting point for the optimization and development of new and promising sEH inhibitors.

## 3. Materials and Methods

### 3.1. Computational Details

#### 3.1.1. Preparation of the Library

Using CombiGlide software (version 4.4), a library of 8772 compounds was generated, considering 2924 commercially available aromatic amines, according to the synthetic route reported in Figure 1. Subsequently, LigPrep was applied for the generation of all possible tautomers, stereoisomers, and protonation states at physiological pH, while QikProp [66,67] (version 5.1, Schrödinger Suite, Schrödinger, Inc., New York, NY, USA) was employed to predict the pharmacokinetic parameters for each item of the libraries. After that, the new library was filtered using LigFilter (KNIME AG, Zurich, Switzerland), according to the Lipinski filter, to prioritize drug-like compounds, and, finally, 3693 compounds were selected for the subsequent molecular docking calculations.

#### 3.1.2. Molecular Docking Experiments on sEH

A 3D protein model was prepared using the Schrödinger Protein Preparation Wizard [68,69], starting from the sEH X-ray structure in the active form co-complexed with the inhibitor BSU (1,3-diphenylurea) (PDB code: 5AI5). The visual inspection of this protein crystal structure revealed that the binding of the co-crystallized inhibitor (BSU) was not assisted by water molecules and, for this reason, we removed them for the subsequent molecular docking experiments. All hydrogens were then added, and bond orders were assigned. The center grid had the coordinates of −16.43 × −11.02 × 15.93 and was characterized by inner and outer box dimensions of 10 × 10 × 10 and 30 × 30 × 30, respectively. The molecular docking experiments on the 190 compounds of the in house-library (Appendix A) were performed using Glide software (version 9.0) [70,71,72,73] and using the Extra Precision (XP) mode, saving 20 maximum poses for each compound for the subsequent analysis. The docking protocol was validated through the redocking of BSU (PDB code: 5AI5, Appendix A).

#### 3.1.3. Development of the 3D Structure-Based Pharmacophore Model for sEH

20 sEH inhibitors [40,41,42,43,44,45,46,47,48,49,50] whose coordinates and information were available in the Protein Data Bank (PDB codes 1EK2, 1VJ5, 3ANS, 3ANT, 3WKE, 4HAI, 4OCZ, 4OD0, 5AI5, 5AK5, 5AKE, 5ALG, 5ALP, 5ALU, 5ALZ, 5AM1, 6AUM, 6FR2, 6HGX and 6YL4 [40,41,42,43,44,45,46,47,48,49]) were used to build structure-based three-dimensional pharmacophore models. In order to generate these models, all the ligands must be in the same coordinates system. For these reasons, a crystal structure of sEH (PDB code: 5AI5) was chosen as the reference protein system for performing the starting molecular docking step (Glide software, version 9.0), accounting for the 20 selected sEH inhibitors as ligand input, to reproduce the original experimental ligands binding modes, as detected by careful visual inspection. Afterwards, the sampled poses were subsequently used as inputs for generating the “structure-based 3D pharmacophore” models through the Develop Pharmacophore Hypothesis panel. The function “use prealigned ligands” was used to preserve the coordinates of the sampled poses. Using the default parameters, i.e., hypotheses must match 50% of the ligands and tolerance set to 2 Å, the generated hypotheses featured only three pharmacophoric points. In our experience, a 3-point pharmacophoric model is not convenient as it is poorly selective and representative of a possible binder. Therefore, we modified the default parameters by accounting at least 25% of the input ligands and setting the tolerance to 2.5 Å; in this way, 5-point structure-based three-dimensional pharmacophore model (AADHR) was generated. This evidence suggests that known co-crystallized ligands of soluble epoxide hydrolase possess such variability that more pharmacophoric models could probably be accounted for with regard to this protein.

Specifically, following the definitions of specific features as implemented in the Develop Pharmacophoric Hypothesis panel (Phase [63,64,65]), “A” indicates an acceptor group, “D” indicates a donor group, “H” a hydrophobic one, and “R” an aromatic ring.

#### 3.1.4. Pharmacophore Screening

Pharmacophore screening was performed before and after molecular docking calculations. Firstly, 190 in-house synthesized compounds were indeed preliminarily screened using the generated pharmacophoric model “pharm-sEH” (AADHR model) and the “Ligand and database screening” tool in Phase [63,64,65]. Specifically, the “generate multiple conformers” option was set, with a maximum of 50 conformers for each molecule, thus performing a conformational search aimed at evaluating the matching with the pharmacophoric features a priori. Subsequently, 89 compounds matching all pharmacophoric points were submitted to molecular docking experiments. The output docking poses were again screened using “pharm-sEH” pharmacophoric model and, in this case, the specific conformer accommodated in the chosen protein structure was taken into account, skipping any further conformational search (i.e., skipping the “generate multiple conformers” option, as reported above). After this step, only five of 89 molecules, i.e., **3**, **4**, **6**, **7**, and **8**, matched all the pharmacophoric points, featuring a phase screen score from 0.84 to 0.63, which is in line with the maximum value (phase screen score = 1.23, obtained for 2RV ligand, PDB code: 4OD0, www.rcsb.org, accessed on: 4 February 2021) obtained after screening all the known sEHi accounted for the pharmacophore model generation.

### 3.2. Chemical Synthesis

All commercially available starting materials were purchased from Merck KGaA (Darmstadt, Germany) and used without further purification. The solvents for the synthesis were of HPLC grade (Merck KGaA Darmstadt, Germany). NMR spectra (^1^H, ^13^C) were recorded on Bruker Avance 400 MHz or 500 MHz instruments (Billerica, MA, USA), T = 298 K. Compounds were dissolved in 0.5 mL of CD_3_OD or CDCl_3_ (Merck KGaA, Darmstadt, Germany) 99.8 Atom %D). Coupling constants (*J*) are reported in Hertz, and chemical shifts are expressed in parts per million (ppm) on the delta (δ) scale relative to the solvent peak as the internal reference. Multiplicities are reported as follows: s, singlet; d, doublet; dd, doublet of doublets; ddd, doublet of doublet of doublets; t, triplet; td, triplet of doublets; p, pentet; m, multiplet. Reactions were monitored on silica gel 60 F254 plates (Merck KGaA, Darmstadt, Germany), and the spots were visualized under UV light (λ = 254 nm, 365 nm). Analytical and semi-preparative reversed-phase HPLC was performed on Agilent Technologies 1200 Series high performance liquid chromatography (Santa Clara, CA, USA) using a Nucleodur (Macherey-Nagel GmbH & Co. KG, Düren, Germany), C8 reversed-phase column (75 × 4.6 mm, 5 μm, 80 Å, flow rate = 1 mL/min; 250 × 10.0 mm, 5 μm, 80 Å, flow rate = 4 mL/min respectively, Macherey Nagel^®^). The binary solvent system (A/B) was as follows: 0.1% TFA in water (A) and 0.1% TFA in CH_3_CN (B). The absorbance was detected at 240 nm. The purity of all tested compounds (>97%) was determined by HPLC analysis.

#### 3.2.1. General Procedure (A) for the Synthesis of **2a**–**2c**

Triethylamine (1.0 equiv.) was added to a solution of **a**–**c** (1.0 equiv.) in water (2.6 mL) followed by a portion wise addition of isatoic anhydride **1** (1.1 equiv.). The reaction mixture was stirred for 2 h at 30–40 °C, cooled to room temperature and evaporated in vacuum to form an oil residue. This material was refluxed for 7 h in formic acid (3.6mL), cooled to room temperature and evaporated. The solid was resuspended in water, extracted with DCM (3 × 25 mL), dried over anhydrous Na_2_SO_4_, filtered, and concentrated. The desired compounds **2a**–**2c** were confirmed by analytical RP-HPLC (Nucleodur, C8 reversed-phase column: 100 × 2 mm, 4 μM, 80 Å, flow rate = 1 mL/min) and used without any further purification for the next step [74].

##### 6-(2,4-dioxo-1,2-dihydroquinazolin-3(4H)-yl)hexanoic acid (**2b**)

Compound **2b** was obtained by following the general procedure (A) as a brown solid (315 mg, 60% yield after HPLC purification). RP-HPLC t_R_ = 17.5 min, gradient condition: from 5% B ending to 100% B 50 min, flow rate of 4 mL/min, λ = 240 nm. ^1^H NMR (400 MHz, CD_3_OD): δ = 8.27 (d, *J* = 8.0 Hz, 1H), 7.85 (t, *J* = 7.7 Hz, 1H), 7.72 (d, *J* = 8.1 Hz, 1H), 7.59 (t, *J* = 7.6 Hz, 1H), 4.09 (t, *J* = 7.3 Hz, 2H), 2.42 (t, *J* = 7.5 Hz, 2H), 1.85 (p, *J* = 7.5 Hz, 2H), 7.69 (p, *J* = 7.6 Hz, 2H), 1.45 (p, *J* = 8.0 Hz, 2H). ^13^C NMR (100 MHz, CD_3_OD): δ = 174.45, 162.06, 148.23, 137.79, 134.86, 125.60, 122.30, 114.88, 112.03, 45.23, 35.82, 28.16, 26.75, 24.93.

#### 3.2.2. General Procedure (B) for the Synthesis of **3**–**8**

In a flask containing **2a**–**2c** (1.0 equiv.) in DMF (2 mL), 1-hydroxybenzotriazole (2.0 equiv.) and *N*,*N*′-diisopropylcarbodiimide (1.5 equiv.) were added. The mixture was left under magnetic stirring at room temperature for 1 h, then aniline **d**–**h** (2.0 equiv.) was added and the reaction stirred for 16 h at room temperature. After completion, the mixture was poured into water and extracted with EtOAc (3 × 25 mL). The combined organic phases were washed with a saturated solution of NaHCO_3,_ brine and finally dried under vacuum. HPLC purification was performed by semi-preparative reversed-phase HPLC (Nucleodur, C8 reversed-phase column: 250 × 10.00 mm, 4 μM, 80 Å, flow rate = 4 mL/min) and the final products were characterized by NMR spectra [75].

##### *N*-(2,3-dihydrobenzo[b][1,4]dioxin-6-yl)-5-(2,4-dioxo-1,2dihydroquinazolin-3(4H) yl)pentanamide **3**

Compound **3** was obtained by following the general procedure (A–B) as a brown solid (310 mg, 20% yield after HPLC purification). RP-HPLC t_R_ = 26.2 min, gradient condition: from 5% B ending to 100% B 50 min, flow rate of 4 mL/min, λ = 240 nm. ^1^H NMR (400 MHz, CDCl_3_): δ = 8.27 (d, *J* = 8.0 Hz, 1H), 7.82–7.74 (m, 2H), 7.58–7.53 (m, 2H), 6.82 (dd, *J* = 8.7, 2.4 Hz, 1H), 6.69 (d, *J* = 8.6, 1H), 4.20–4.16 (m, 4H), 4.08 (t, *J* = 7.1 Hz, 2H), 2.38 (t, *J* = 7.2 Hz, 2H), 1.86 (p, *J* = 7.2 Hz, 2H), 1.75 (p, *J* = 7.3 Hz, 2H). ^13^C NMR (100 MHz, CD_3_OD): δ = 173.71, 162.64, 149.57, 148.44, 144.86, 141.93, 135.99, 133.57, 128.97, 127.79, 127.41, 122.97, 118.04, 114.73, 110.94, 65.85, 65.65, 48.04, 37.23, 29.87, 23.92.

##### *N*-(2,3-dihydrobenzo[b][1,4]dioxin-6-yl)-6-(2,4-dioxo-1,2-dihydroquinazolin-3(4H)-yl)hexanamide **4**

Compound **4** was obtained by following the general procedure (A–B) as a brown solid (130 mg, 22% yield after HPLC purification). RP-HPLC t_R_ = 23.6 min, gradient condition: from 5% B ending to 100% B 50 min, flow rate of 4 mL/min, λ = 240 nm. ^1^H NMR (400 MHz, CDCl_3_): δ = 8.33 (dd, *J* = 8.0, 1.4 Hz, 1H), 8.14 (s, 1H), 7.82–7.73 (m, 2H), 7.57–7.52 (m, 1H), 7.17 (d, *J* = 2.5 Hz, 1H), 6.91 (dd, *J* = 8.7, 2.5 Hz, 1H), 6.80 (d, *J* = 8.7 Hz, 1H), 4.25 (s, 4H), 4.04 (t, *J* = 7.4 Hz, 2H), 2.36 (t, *J* = 7.4 Hz, 2H), 1.93–1.76 (m, 4H), 1.49 (p, *J* = 7.7 Hz, 2H). ^13^C NMR (100 MHz, CDCl_3_): δ = 170.71, 160.93, 147.50, 146.76, 143.46, 140.42, 134.43, 131.63, 127.50, 127.04, 126.80, 121.96, 117.12, 113.54, 109.76, 64.43, 64.28, 46.86, 37.00, 28.86, 25.95, 24.75.

##### *N*-(5-bromopyridin-3-yl)-5-(2,4-dioxo-1,2-dihydroquinazolin-3(4H)-yl)pentanamide **5**

Compound **5** was obtained by following the general procedure (A–B) as a brown solid (235 mg, 25% yield after HPLC purification). RP-HPLC t_R_ = 21.3 min, gradient condition: from 5% B ending to 100% B 50 min, flow rate of 4 mL/min, λ = 240 nm. ^1^H NMR (400 MHz, CD_3_OD): δ = 8.70 (s, 1H), 8.59 (s, 1H), 8.45 (s, 1H), 8.39 (s, 1H), 8.29 (dd, *J* = 8.1, 1.4 Hz, 1H), 7.89 (ddd, *J* = 8.5, 7.2, 1.5 Hz, 1H), 7.73 (d, *J* = 8.2 Hz, 1H), 7.63 (t, *J* = 7.6 Hz, 1H), 4.16 (t, *J* = 7.1 Hz, 2H), 2.53 (t, *J* = 7.1 Hz, 2H), 1.92 (p, *J* = 7.3 Hz, 2H), 1.82 (p, *J* = 7.3Hz, 2H). ^13^C NMR (100 MHz, CD_3_OD): δ = 172.97, 160.90, 148.19, 146.32, 143.91, 138.03, 137.02, 134.59, 131.30, 129.70, 127.49, 126.14, 125.46, 121.38, 46.53, 35.29, 28.17, 21.82.

##### *N*-(5-bromopyridin-3-yl)-7-(2,4-dioxo-1,2-dihydroquinazolin-3(4H)-yl)heptanamide **6**

Compound **6** was obtained by following the general procedure (A–B) as a brown solid (150 mg, 35% yield after HPLC purification). RP-HPLC t_R_ = 22.8 min, gradient condition: from 5% B ending to 100% B 50 min, flow rate of 4 mL/min, λ = 240 nm. ^1^H NMR (400 MHz, CD_3_OD): δ = 8.66 (s, 1H), 8.48–8.44 (m, 2H), 8.38 (s, 1H), 8.28 (dd, *J* = 8.0, 4.2 Hz, 1H), 7.88 (t, *J* = 7.6 Hz, 1H), 7.73 (dd, *J* = 8.1, 2.7 Hz, 1H), 7.61 (td, *J* = 7.7, 2.9 Hz, 1H), 4.13–4.07 (m, 2H), 2.35 (t, *J* = 7.4 Hz, 2H), 1.84 (p, *J* = 7.7 Hz, 2H), 1.68–1.63 (m, 2H), 1.50–1.41 (m, 4H). ^13^C NMR (125 MHz, CD_3_OD): δ = 173.61, 160.86, 148.01, 146.68, 143.96, 138.25, 137.05, 134.43, 129.65, 128.71, 127.39, 126.20, 125.73, 121.45, 46.83, 36.23, 28.58, 28.29, 25.88, 24.83.

##### 6-(2,4-dioxo-1,2-dihydroquinazolin-3(4H)-yl)-*N*-(6-iodopyridin-3-yl)hexanamide **7**

Compound **7** was obtained by following the general procedure (A–B) as a brown solid (240 mg, 20% yield after HPLC purification). RP-HPLC t_R_ = 22.7 min, gradient condition: from 5% B ending to 100% B 50 min, flow rate of 4 mL/min, λ = 240 nm. ^1^H NMR (400 MHz, CD_3_OD): δ = 8.59–8.55 (m, 2H), 8.26 (dd, *J* = 8.1, 1.6 Hz, 1H), 7.91–7.87 (m, 1H), 7.80–7.75 (m, 2H), 7.71 (d, *J* = 8.1 Hz, 1H), 7.62 (t, *J* = 7.7 Hz, 1H), 4.13 (t, *J* = 7.3 Hz, 2H), 2.44 (t, *J* = 7.3 Hz, 2H), 1.88 (p, *J* = 7.4 Hz, 2H), 1.78 (p, *J* = 7.3 Hz, 2H), 1.52–1.44 (m, 2H). ^13^C NMR (100 MHz, CD_3_OD): δ = 173.29, 161.13, 147.98, 146.95, 141.63, 135.95, 134.70, 134.37, 128.93, 127.28, 126.12, 126.03, 121.38, 108.26, 46.61, 35.97, 28.38, 26.63, 24.59.

##### *N*-(benzo[d]thiazol-6-yl)-6-(2,4-dioxo-1,2-dihydroquinazolin-3(4H)-yl)hexanamide **8**

Compound **8** was obtained by following the general procedure (A–B) as a brown solid (147 mg, 20% yield after HPLC purification). RP-HPLC t_R_ = 20.9 min, gradient condition: from 5% B ending to 100% B 50 min, flow rate of 4 mL/min, λ = 240 nm. ^1^H NMR (400 MHz, CDCl_3_): δ = 8.94 (s, 1H), 8.36 (d, *J* = 10.1 Hz, 1H), 8.28 (dd, *J* = 6.2, 3.4 Hz, 1H), 8.11 (d, *J* = 8.3 Hz, 1H), 8.01 (d, *J* = 8.3 Hz, 1H), 7.98–7.92 (m, 1H), 7.85 (d, *J* = 7.7 Hz, 1H), 7.64 (t, *J* = 7.7 Hz, 1H), 4.33–4.20 (m, 2H), 2.58–2.46 (m, 2H), 2.00–1.78 (m, 4H), 1.59–1.45 (m, 2H). ^13^C NMR (100 MHz, CD_3_OD): δ = 172.91, 160.84, 154.47, 149.05, 148.10, 146.63, 136.61, 134.45, 134.19, 127.31, 126.09, 125.76, 122.32, 121.37, 119.27, 112.23, 46.65, 36.13, 28.44, 25.61, 24.78.

### 3.3. SPR Assays on Hsp90

Surface plasmon resonance (SPR) analyses were performed to determine the binding of **2b** and **3**–**8** to full-length Hsp90α using a Biacore 3000 (Cytiva, Marlborough, MA, USA) equipped with research-grade CM5 sensor chips (GE Healthcare). Recombinant human Hsp90α was purchased from Abcam (Abcam, Cambridge, UK). The protein was coupled to the surface of a CM5 sensor chip using standard amine-coupling protocols according to the manufacturer’s instructions. One unmodified reference surface was prepared for simultaneous analyses. Hsp90α (100 μg mL^−1^ in 10 mM CH_3_COONa, pH 4.5) was immobilized on an individual sensor chip surface at a flow rate of 5 μL min^−1^ to obtain densities of 11–12 kRU. Compounds **2b**, **3**–**8** were dissolved to obtain 40 mM solutions in 100% DMSO and diluted 1:100 (*v*/*v*) in PBS (10 mM NaH_2_PO_4_, 150 mM NaCl, pH 7.4) to a final DMSO concentration of 1.0%. For each molecule, a six-point concentration series was set up, spanning 0, 0.02, 0.08, 0.25, 1.0 and 4.0 μM, and for each sample, the complete binding study was performed using triplicate aliquots. SPR experiments were performed at 25 °C, using a flow rate of 10 µL min^−1^, with 60 s monitoring of association and 300 s monitoring of dissociation. Changes in mass, due to the binding response, were recorded as resonance units (RU). To obtain the dissociation constant (K_D_), these responses were fit to a 1:1 Langmuir binding model by nonlinear regression using the BiaEvaluation software program (version 4.1, Biacore AB, Uppsala, Sweden) provided by GE Healthcare. Simple interactions were suitably fitted to a single-site bimolecular interaction model (A + B = AB), yielding a single K_D_. No binding was observed for any of the tested molecules, **3**–**8**, against Hsp90α.

### 3.4. Biological Evalutaion on sEH

#### 3.4.1. Expression, Purification, and Activity Assay of Human Recombinant sEH

Human recombinant sEH was expressed and purified as reported before [76]. In brief, Sf9 cells were infected with a recombinant baculovirus, kindly provided by Dr. B. Hammock, University of California, Davis, CA, USA. After 72 h, cells were pelleted and sonicated (3 × 10 s at 4 °C) in a lysis buffer containing NaHPO_4_ (50 mM, pH 8), NaCl (300 mM), glycerol (10%), EDTA (1 mM), phenyl-methanesulphonylfluoride (1 mM), leupeptin (10 mg/mL), and soybean trypsin inhibitor (60 mg/mL). A centrifugation step (100,000× *g*, 60 min, 4 °C) was applied, and supernatants were collected and subjected to benzyl-thiosepharose-affinity chromatography to purify sEH by elution with 4-fluorochalcone oxide in PBS containing DTT (1 mM) and EDTA (1 mM). A dialyzed and concentrated (Millipore Amicon-Ultra-15 centrifugal filter) enzyme solution was assayed for total protein with a Bio-Rad protein detection kit (Bio-Rad Laboratories, Munich, Germany), and the activity of sEH was determined by using a fluorescence-based assay as described before.

For the evaluation of the activity of test compounds (**3**–**8**), purified sEH was diluted in a Tris buffer (25 mM, pH 7) supplemented with BSA (0.1 mg/mL) to an appropriate enzyme concentration (depending on the precedent measured activity) and pre-incubated with compounds **3**–**8** at 10 μM or vehicle (0.1% DMSO) for 15 min at room temperature. The reaction was started by the addition of 50 μM 3-phenyl-cyano(6-methoxy-2-naphthalenyl)methyl ester-2-oxiraneacetic acid (PHOME), a non-fluorescent compound that is enzymatically converted into fluorescent 6-methoxy-naphtaldehyde at rt. The reaction was stopped after 1 h by ZnSO_4_ (200 μM) and fluorescence was detected (λ_em_ = 465 nm, λ_ex_ = 330 nm).

#### 3.4.2. Cell Viability Assay on PBMC

PBMC were treated with the indicated compounds (1 or 10 µM) or toxic controls (50 nM triptolide or 0.0125% Triton-X) for 24 h. Cell viability was assessed by adding 20 µL of a solution of 3-(4,5-dimethylthiazol-2-yl)-2,5-diphenyltetrazolium bromide (MTT, 5 mg/mL; Sigma-Aldrich, Munich, Germany) per 100 µL sample suspension and incubating for another 3 h at 37 °C and 5% CO_2_. Formazan was solubilized by adding 100 µL of SDS solution (10% in in 20 mM HCl) and shaking for 20 h in the dark. The absorbance at 570 nm was measured using a Multiskan Spectrum microplate reader (Thermo Fischer Scientific, Schwerte, Germany). Viability (%) was calculated by comparing the absorbance of samples to that of vehicle controls. Statistical testing was performed by one-way ANOVA on raw absorbance without correction but yielded no significant differences for compounds **3**–**8**.

## 4. Conclusions

In conclusion, we have demonstrated that the developed sEH 3D structure-based pharmacophore model is a new interesting computational tool to accelerate and assist the drug discovery process, not only in the design and development of new bioactive molecular platforms but also in drug repositioning campaigns. In this study, we performed target identification through “pharm-sEH”: precisely, a repositioning has been implemented, starting from an in-house library for organic synthetic compounds **3**–**8**, initially designed for Hsp90, which otherwise would have been synthesized and discarded for further investigation. Through a precise computational workflow, which includes pharmacophore screening before and after molecular docking calculation, compounds **3**–**8** were investigated in a targeted fashion and supported by computational predictions on sEH. Biological results corroborated the preliminary data since compounds **3** and **4** were identified as promising bioactive compounds on sEH (IC_50_ **3** = 8.8 ± 1.5 µM and **4** = 4.5 ± 1.0 µM) for the treatment of inflammation process. The present outcomes also suggest further investigation of the 1,4-benzodioxane scaffold for identifying new promising sEH inhibitors since it is shared by the two active compounds and an already known inhibitor (i.e., R4N ligand, reference PDB code:5ALG).

The main outcome of this work is represented by the development and validation of the 3D structure-based pharmacophore model “pharm-sEH” that highlighted the structural determinants responsible for the sEH binding. Notably, it could also be useful in accelerating the future design and identification of novel sEH inhibitors and the reinvestigation of shelved compounds, as reported in this case study. Finally, these outcomes pointed out the efficiency of the straightforward reiterable methodology supported by this novel tool. A 3D structure-based pharmacophore model thus constitutes one of the newest attractive computational methodologies to support the drug discovery process and even more drug repositioning campaigns.

## Data Availability

All the relevant data are presented within the body of this paper and Appendix A.

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
