# Peer review of "Repositioning of Quinazolinedione-Based Compounds on Soluble Epoxide Hydrolase (sEH) through 3D Structure-Based Pharmacophore Model-Driven Investigation"

_molecules, 2022, doi:10.3390/molecules27123866_

Round 1

Reviewer 1 Report

Repositioning of Quinazolinedione-Based Compounds on Soluble Epoxide Hydrolase (sEH) through 3D Structure-Based Pharmacophore Model-Driven Investigation

The manuscript (MS) entitled ‘Repositioning of Quinazolinedione-Based Compounds on Soluble Epoxide Hydrolase (sEH) through 3D Structure-Based Pharmacophore Model-Driven Investigation' by Erica Gazzillo et al. presents combined computational and experimental study of quinazolinedione-based compounds as potential soluble epoxide hydrolase inhibitors.

  1. The introduction should be expanded. What is the current status of research concerning sEH inhibitors? Are there other studies exploiting 3D structure-based pharmacophoric models in computationally aided drug design? What are benefits and potential problems?
  2. What was the motivation to design model for sEH, after compounds turned out not to be inhibitors of His90?
  3. General workflow (Figure 2) is confusing. Did not the Authors select target at the beginning, and not after experimental binding on sEH was confirmed?
  4. Could the Authors please comment the failure of docking approach for 3-8 binding potential towards Hsp90? Is it due to the poor performance of the docking algorithm or due to the (only) 60% similarity between yeast Hsp82 and human Hsp90?
  5. Why the Authors did not try to model human Hsp90 using AlphaFold, for example, or some homology modeling tool?
  6. It would be useful if the Authors would add IC50 values of co-crystal structures to Figure 5, next to their PDB IDs.
  7. What is the IC50 value for the inhibitor R4N? Is it better inhibitor than compounds 3 and 4?
  8. The validation of the docking protocol is missing.
  9. Could the Authors please comment the toxicity of their proposed inhibitors? Are the proposed compounds ‘only’ inhibitors or possible drug candidates?
  10. What is the main result of the present study? Synthesis? Methodology? New inhibitors with better properties compared to existing ones?

Reviewer 2 Report

This is a good manuscript, generally wel-writen in spite of occasional language deficiencies.  They describe the discovery of two moleucles with IC50 under 10 uM against soluble epoxide hydrolase, which may be good scaffolds for the development of additional inhibitors. I support its publication after the few minor points below are addressed:

in line 100, authors state "After applying different filters.." Please describe the filters thoroughly

lines 261-264: authors say used Hsp82 structure. "The Hsp82 X-ray structure was chosen for molecular docking experiments due to the 261 lack of Hsp90 crystal structures," However, they state later that a comparison between hsp82 and hsp90 has a 0.91 angstrom RMSD, which implies that partial Hsp90 structures are indeed available, and a search of the PDB indeed finds hsp90 structures: 2XJX, 2XDL, 3FT5, 3HYZ ...  The rational behind the choice of hsp82 instad of hsp90 must be much better explained. For example: is the structure of the mechanistically relevant portion of the protein absent for hsp90 and present for hsp82? etc. Also: why did authors not attempt to dock against an homology model of hsp90 built from the hsp82 template?

section 2.3.  Authors do not quite explain why they decided to try to reposition their failed hsp90 hits against sEH, rather than any of a dozen other targets. The reader would benefit from a discussion of the rationale behind that decision: was it deliberate or a fortuitous positive result after trying to reposition them against a myriad of targets?

Round 2

Reviewer 1 Report

I suggest to publish as it is.